

# Isolation and characterization of native antagonistic rhizobacteria against Fusarium wilt of chilli to promote plant growth

Bhanothu Shiva[1], Petikam Srinivas[2], Deepa Khulbe[1], Lellapalli Rithesh[3], Penumatsa Kishore Varma[4], Rahul Kumar Tiwari[5], Milan Kumar Lal[5] and Ravinder Kumar[6]

[1] Department of Plant Pathology, College of Agriculture, Odisha University of Agriculture and Technology, Bhubaneswar, Odisha, India
[2] Central Horticultural Experiment Station, Indian Council of Agricultural Research-Indian Institute of Horticultural Research, Bhubaneswar, Odisha, India
[3] Department of Plant Pathology, Kerala Agricultural University, Thiruvanathapuram, Kerala, India
[4] Plant Protection Division, Regional Agricultural Research Station, Acharya Ranga Agricultural University, Lam, Telangana, India
[5] Division of Plant Protection, ICAR-Central Potato Research Institute, Shimla, Himachal Pradesh, India
[6] Division of Plant Pathology, ICAR-Indian Agricultural Research Institute, New Delhi, Delhi, India

Corresponding authors
Deepa Khulbe,
petikamdeepa@gmail.com
Ravinder Kumar,
chauhanravinder97@gmail.com

## ABSTRACT

In the eastern coastal regions of Odisha, wilt caused by *Fusarium oxysporum* f. sp. *capsici* is an extremely damaging disease in chilli. This disease is very difficult to manage with chemical fungicides since it is soil-borne in nature. The natural rhizosphere soil of the chilli plant was used to isolate and test bacterial antagonists for their effectiveness and ability to promote plant growth. Out of the fifty-five isolates isolated from the rhizosphere of healthy chilli plants, five isolates, namely Iso 01, Iso 17, Iso 23, Iso 24, and Iso 32, showed their highly antagonistic activity against *F. oxysporum* f. sp. *capsici* under *in vitro*. In a dual culture, Iso 32 (73.3%) and Iso 24 (71.5%) caused the highest level of pathogen inhibition. In greenhouse trials, artificially inoculated chilli plants treated with Iso 32 (8.8%) and Iso 24 (10.2%) had decreased percent disease incidence (PDI), with percent disease reduction over control of 85.6% and 83.3%, respectively. Iso 32 and Iso 24 treated chilli seeds have shown higher seed vigor index of 973.7 and 948.8, respectively, as compared to untreated control 636.5. Furthermore, both the isolates significantly increased plant height as well as the fresh and dry weight of chilli plants under the rolled paper towel method. Morphological, biochemical, and molecular characterization identified *Bacillus amyloliquefaciens* (MH491049) as the key antagonist. This study demonstrates that rhizobacteria, specifically Iso 32 and Iso 24, can effectively protect chilli plants against Fusarium wilt while promoting overall plant development. These findings hold promise for sustainable and eco-friendly management of Fusarium wilt in chilli cultivation.

# INTRODUCTION

Chilli (*Capsicum annuum* L.) is a significant crop of vegetables and spices with medicinal and nutritional properties. Fresh green fruits are used in vegetables and salads, while ripe red fruits are used as condiments, digestive aids, and flavoring and coloring ingredients in sauces, chutneys, pickles, and other culinary preparations (*Welbaum, 2015*). Chilli is rich in vitamins A and C, iron, potassium, and magnesium, and it can boost the immune system and reduce cholesterol levels (*Grubben & Mohamed El, 2004*). Chilli is a major tropical and subtropical crop cultivated on 19.89 million hectares worldwide (*Food & Agriculture Organization of the United Nations, 2020*). According to *Indian Horticulture Database (2022)*, India was the world's largest producer, consumer, and exporter of dried chilli, with 8.52 lakh hectares and an output of 15.78 lakh metric tons. China, Mexico, Peru, and other countries were next in line. Many fungal, bacterial and viral infections affect chillies, including soil-borne disease wilt caused by *Fusarium oxysporum* (Schlect.) emend. Synd. & Hans. f.sp. *capsici* Riv. may result in production losses of up to 50% (*Singh & Srivastava, 1953*). High humidity and high temperatures significantly affect the development of symptoms because the pathogen is soil-borne in nature (*Sonago, 2003*). The pathogen invades roots, multiplies, spreads in the root and stem vascular systems, and blocks the circulation of water and nutrients (*Miller, Rowe & Riedel, 1986*). Recent assessments in the coastal zone of Odisha's chilli-growing regions revealed that, in addition to chilli leaf curl disease, wilt disease is also heavily affecting yields (*Sarkar et al., 2018*).

Historically, synthetic fungicides have been applied to manage this disease. In addition to the pathogens developing resistance to fungicides, widespread usage of fungicides caused the buildup of toxic residues, ecosystem pollution, and disturbed the soil's biological equilibrium by eradicating the non-targeted microbes (*Sela-Buurlage et al., 2001*). Therefore, it is crucial to provide a non-chemical treatment for the condition that is efficient, affordable, and safe for the environment (*Apastambh, Tanveer & Baig, 2016*). Consequently, biological control has been developed as an alternative to chemical fungicides by employing antagonistic microorganisms for disease control, with remarkable success (*Landa, Navas- Cortés & Jiménez-Dìa, 2004*). However, biological control using plant growth-promoting rhizobacteria (PGPR) represents a cost-effective, environmentally friendly, and reliable approach (*Compant et al., 2005*). Notably, PGPR strains, including those belonging to *Bacillus, Enterobacter*, and *Pseudomonas* genera, emerge as key root colonizers. They play a pivotal role in enhancing plant defenses by inducing systemic resistance (ISR) (*Joseph, Patra & Lawrence, 2007*). *Adekunle et al. (2001)* reported effective management of Fusarium wilt in vegetables through the application of *Pseudomonas* spp., *Bacillus* spp., and *Trichoderma* spp. Compared to chemical fungicides, PGPR offers defense against a range of soil-borne fungal infections while leaving behind no hazardous aftereffects, as highlighted by *Tiwari & Mukhopadhyay (2001)*. Therefore, the current investigation aimed to identify potent native antagonistic PGPR with the capability to

effectively control wilt caused by *F. oxysporum* f. sp. *capsici*, while also promoting plant growth.

# MATERIALS AND METHODS

The study was conducted during 2021–22 at the Department of Plant Pathology, Orissa University of Agriculture and Technology (OUAT), and the ICAR-IIHR Central Horticultural Experiment Station in Bhubaneswar (CHES), Odisha.

## Isolation, characterization and pathogenicity of the wilt pathogen

The pathogen was isolated from wilted plants (Fig. 1) collected through field surveys from the OUAT (20°15′N, 85° 48′E), the Regional Research and Technology Transfer Station (20°15′N, 85°47′E), ICAR-IIHR-CHES (20°14′N, 85°46′E), and local farmer's fields at Balakatti (20°12′N, 85°52′E), and Uttara chowk (20°11′N, 85°52′E). The pathogen associated with the disease was isolated in a pure form on potato dextrose agar (PDA) and identified based on morphological traits, including fungal culture characteristics, microscopic features, and conidia shape (*Booth, 1971*). A pathogenicity test was then conducted to assess the ability of the *F. oxysporum* f. sp. *capsici* isolate to induce typical disease symptoms under artificial conditions, using the chilli variety Agni Jwala. The pathogen was inoculated into powdered maize grain seeds for mass multiplication, added to wet coir pith at a rate of 10 g kg$^{-1}$, and properly mixed. Without adding inoculum to the coir pith, an adequate check was maintained. After 13 days, the seedlings were checked for the emergence of symptoms. To validate the identification and pathogenicity, *F. oxysporum* f. sp. *capsici* was re-isolated from infected seedlings, and the cultures obtained from infected seedlings and compared with previously isolated pure cultures.

## Isolation and characterization of native PGPR

Isolation of native PGPR from collected soil samples was carried out by the dilution plate technique as described by *King, Ward & Raney (1954)* on King's B agar medium (KB) and incubated at 27 °C for 2–4 days. The isolates were characterized for carbohydrate utilization using HiCarbo Kits (Part B and Part C) from Hi Media Pvt Limited, Mumbai. Part B contains 12 wells for carbohydrate utilization tests, while Part C contains 11 wells for sugars and 1 well for control. Each well was stab inoculated with a loopful of bacterial culture and incubated at 25 ± 2 °C for 16 h. Bacterial reactions to carbohydrate utilization were recorded 4 days after incubation, interpreting results based on medium colour changes. Data analysis was performed using "ABIS Online," an online program for bacterial identification.

### Molecular characterization

The 16S rRNA gene was amplified from chosen isolate Iso 32 using universal primers 27F (5′-AGAGTTTGATCCTGGCTCAG-3′) and 1492R (5-GGTTACCTTGTTACGACTT-3′) (*Frank et al. 2008*) and sequenced (Eurofins, bangalore, Luxembourg). The PCR reaction mixture consisted of 20 µl of master mix (Takara emerald master mix), 8 µl sterile distilled water, 4 µl each of 10 pM primers, and 4 µl of genomic DNA for a total volume of 40 µl. PCR amplification conditions were; as follows initial denaturation: 95 °C for 4 min;

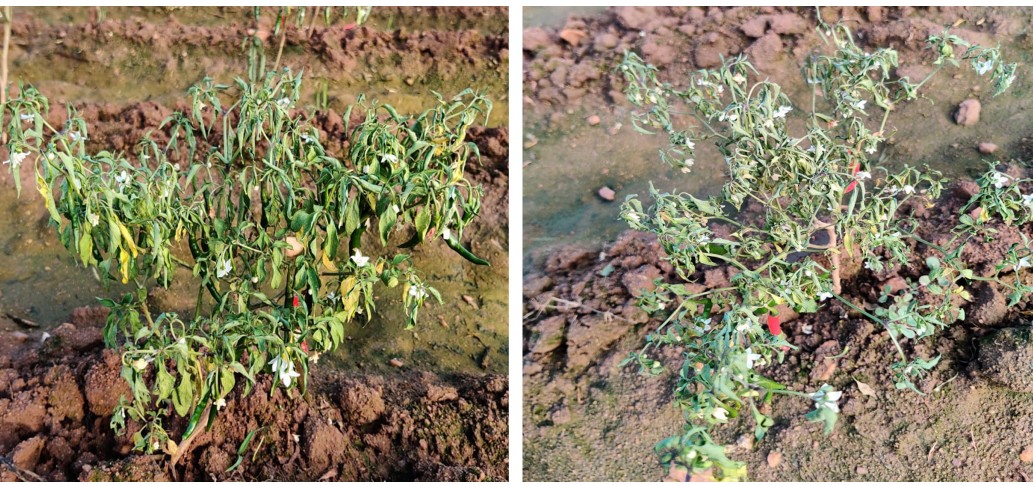

**Figure 1 Chilli plants showing severe wilting symptoms.**

denaturation: 95 °C for 60 s; annealing: 56 °C for 60 s 34 cycles; extension: 72 °C for 1 min; final extension: 72 °C for 10 min. PCR amplification products were examined by preparing 2% agarose gel using 0.5X TAE buffer which is stained in ethidium bromide solution. The PCR amplified products with a DNA ladder run at 80 V for 2 h in an electrophoresis unit. Under UV trans-illuminator gel profile was examined and acknowledged by the use of a gel documentation unit.

The resulting partial 16S rRNA gene sequences (varying from 400 to 1,200 nucleotides) were compared to the sequences in the National Center for Biotechnology Information (NCBI) nucleotide database. The evolution history was inferred using the Neighbor-Joining method (*Saitou & Nei, 1987*). The bootstrap consensus tree inferred from 1,000 replicates is taken to represent the evolutionary history of the taxa analyzed (*Felsenstein, 1985*). Branches corresponding to partitions reproduced in less than 50% of bootstrap replicates are collapsed. The evolutionary distances were computed using the maximum composite likelihood method and are in the units of the number of base substitutions per site (*Tamura, Nei & Kumar, 2004*). This analysis involved 15 nucleotide sequences. All ambiguous positions were removed for each sequence pair (pairwise deletion option). There were a total of 1,584 positions in the final dataset. Evolutionary analyses were conducted in MEGA11 (*Tamura, Stecher & Kumar, 2021*).

## HCN production

The isolate's HCN generation was evaluated using *Bakker & Schipper*'s *(1987)* assay. Individual cultures of antagonistic bacteria were inoculated into KB agar medium supplemented with 4.4 g/L glycine, and to provide a comparison, uninoculated controls were utilized. Each Petri dish cover was filled with filter paper discs (Whatman No. 1) soaked in 0.5% picric acid and 2% sodium carbonate. After being parafilm-sealed, the plates were incubated for 4 days at 25 °C. When the color changed from yellow to reddish brown, the bacteria were producing HCN at moderate and high levels, respectively.

## Phosphate solubilization ability

The solubility of inorganic phosphate (tricalcium phosphate) in bacterial isolates was tested using King's technique (*King, 1936*). After spreading a loopful of fresh bacterial culture over Pikovskaya's (PKV) agar medium that had been supplemented with inorganic phosphate, the plates were placed in an incubator at a temperature of 28 °C for 4 days. Mineral phosphate has been solubilized, as demonstrated by a distinct halo zone around the bacterial colony, and the solubilization efficiency and phosphate solubilization index (PSI) will be calculated (*Edi Premono, Moawad & Vlek, 1996*).

$$\text{Solubilization Efficiency (\%)} = \frac{\text{Solubilization diameter (mm)}}{\text{Colony diameter (mm)}} \times 100$$

$$\text{PSI (Phosphate solubilization index)} = \frac{\text{Zone size}}{\text{Colony size}}$$

## *In vitro* evaluation of rhizobacterial strains against pathogen by dual culture method

The antagonistic potential of native rhizobacterial isolates against the *F. oxysporum* f. sp. *capsici* was examined using a dual culture approach (*Idris, Labuschagne & Korsten, 2007*). The extent of rhizobacterial antagonistic activity against *F. oxysporum* f. sp. *capsici* relative to the control was determined by using percent inhibition (*Vincent, 1927*).

$$\text{Percentage inhibition} = \frac{\text{Control} - \text{Test}}{\text{Control}} \times 100$$

## Evaluation of antagonistic rhizobacteria against *Fusarium* wilt using seed treatment and seedling root dip techniques

### Preparation of pathogen and rhizobacterial inoculum for in vivo screening

The *F. oxysporum* f. sp. *capsici* isolates were multiplied on ground maize grains. This was placed into 1,000 ml flasks at a weight of 400 g and autoclaved twice for 20 min at a pressure of 15 lb psi. The flasks were brought to room temperature before being inoculated with 5 mm-sized discs of a *Fusarium* culture that was actively growing on potato dextrose agar (PDA). Seven discs per flask were added, and the flasks were incubated for 3 weeks at 28 ± 2 °C culture grown on PDA. Each flask contained seven discs, and each was incubated at 28 °C for 3 weeks. The Rhizobacteria inoculum was prepared using the procedures described by *Nandakumar et al. (2001)*. Five chosen antagonist isolates were cultured for 24 h in Kings B broth at 37 °C (109 cfu ml$^{-1}$) and 120 rpm on a rotatory shaker. The broth was applied to chilli seedlings at the 4–5 leaf stage for seedling dipping and soaking.

### Seed treatment and seedling root dip techniques

Five rhizobacterial isolates were chosen, and their capacity to suppress wilt disease on chilli plants (cv. Agni Jwala) was assessed *in vivo* (*Karimi et al., 2012*). *F. oxysporum* f. sp. *capsici* culture was inoculated into trays of regular soil mixture and ground maize grains. After 1 week and 1 day before transplanting the seedlings, 5 ml per well of antagonists containing

109 cfu ml$^{-1}$ were added to the coir pith. Three-week-old chilli seedlings were root-dipped for 45 min in a pathogen-antagonist mixture of coir pith before being transplanted into a bacterial solution of antagonistic bacteria (109 cfu ml$^{-1}$) (*Lemessa & Zeller, 2007*). The seedlings were kept in a greenhouse between 24–28 °C, with a relative humidity of 75% to 90%. When required, sterile water was used to water the seedlings. Wilting signs in plants were used to diagnose the disease.

$$\text{Disease incidence} = \frac{\text{Number of diseased plants}}{\text{Total number of plants observed}} \times 100$$

## Effect of seed treatment with antagonistic rhizobacteria on plant growth response assessed by rolled paper towel and root dip treatment methods

Selective isolated rhizobacteria were tested to see how they affected plant growth when seeds were treated using the rolled towel paper method *in vitro* and the root dip method in pot culture under greenhouse conditions.

### Rolled paper towel method

One hundred seeds, each treated with a different chosen strain, were collected at random from each treatment's four replications and evenly distributed between two wet germination roll paper towels. The towels were rolled, and two rubber bands were used to secure the ends. Thereafter, under a typical 12/12 cycle of light and dark, the rolled sheets holding the seeds were left in an upright posture for 7–10 days at room temperature. After incubation, the shoot and root sections were wiped dry using fine tissue paper, and a new weight was recorded. From the stem's base to the developing tip of the newest leaf, the length of the shoot was measured. Similarly, the length of the root was measured from its origin to the tip of the furthest accessible lateral root. The following formulas were used to calculate the germination percentage and seedling vigour index.

$$\text{Germination percentage (\%)} = \frac{\text{Number of seeds germinated}}{\text{Total number of seeds}} \times 100$$

The seedling vigour index was calculated by using the formula as described by *Baki & Anderson (1972)*: Seedling Vigo. Index = (Root length + Shoot length) × Seed germination (%).

### Root dip treatment

A greenhouse pot experiment was conducted to assess the impact of five rhizobacterial isolates on plant growth parameters using the seedling root dip method. The seedling root dip treatment followed the procedure outlined by *Srinivasan, Gilardi & Garibaldi (2009)*. Chilli seedlings were raised in protrays. Seedlings that were 2 weeks old were removed, and their roots were carefully cleaned before being placed in the appropriate rhizobacterial culture broth for 10 min. After treatment, the seedlings were raised in pots in a greenhouse environment. After 2 weeks, the stimulation of plant growth was evaluated in terms of

growth parameters such as root and shoot length, fresh root and shoot weight, dry root and dry shot weight, and fresh root and shoot weight.

## Data analysis

The data were statistically analyzed by analysis of variance using Grapes 1.1.0 software (*Gopinath et al., 2020*). All the experiments were carried out in triplicates. Results were analyzed using an appropriate analysis programme (*Panse & Sukhatme, 1989*).

## RESULTS

### Isolation, characterization and pathogenicity of wilt pathogen

*F. oxysporum* f. sp. *capsici*, a soilborne pathogen, was isolated from wilted chilli plant samples obtained during the survey (Fig. 1). The fungus exhibited colonies with varying colours - white, pink, salmon, or grey - with velvety to cottony surfaces that underwent colour changes upon spore generation (Fig. 2). Microscopically, the hyphae of *F. oxysporum* f. sp. *capsici* were observed to be filaments, septate, hyaline, and branched at acute or right angles. Notably, *F. oxysporum* f. sp. *capsici* is characterized by the formation of both macroconidia and microconidia (Fig. 2). The *F. oxysporum* f. sp. *capsici* isolate's pathogenicity was evaluated *in vitro* using seedlings. Symptoms, including the collapse of entire seedlings and wilting of the petiole, rachis, and leaflets, were observed 15 days post-inoculation. The leaves gradually changed colour, becoming straw-coloured, yellow, and light brown. The signs of plant wilting in the protrays that had been infected with *F. oxysporum* f. sp. *capsici* resembled those of wilting plants in the main field. To identify the pathogen and assess its pathogenicity, isolates of *F. oxysporum* f. sp. *capsici* were obtained from infected seedlings and compared with previously isolated pure cultures.

### Isolation and characterization of native rhizobacteria

A preliminary *in vitro* dual culture bioassay approach was done on isolated rhizobacterial isolates against *F. oxysporum* f. sp. *capsici*. The severity of antagonism by different isolates against pathogens was measured as a percentage inhibition of mycelial growth. The most effective isolates for preventing the pathogen's mycelial development were isolates 01, 17, 23, 24, and 32, with 55% inhibition or greater, while other strains performed ineffectively (Fig. 3). Based on visual assessments of colony colour and type, isolates Iso 01, Iso 24, and Iso 32 had white colonies, whereas Iso 17 and Iso 23 had light yellow colonies. In the case of colony type, isolates Iso 24 and Iso 32 showed irregular colonies, and isolates Iso 01, Iso 17, and Iso 23 showed round colonies.

Out of five selected rhizobacterial isolates screened for antagonistic ability, disease inhibition and plant growth response by two isolates *viz.*, Iso 24 and Iso 32 were found superior over other isolates and were subjected to further characterization at biochemical (Table 1) and molecular level to use these isolates for prospective application as superior rhizobacterial isolates.

The isolate Iso 24 utilized rhamnose, but not Iso 32. Both tested isolates utilized O-Nitrophenyl β-D galactopyranoside (ONPG), esculin, citrate and malonate remaining sugars are not utilized in Part C. The carbon sources tested, in Part B most of the sugars are

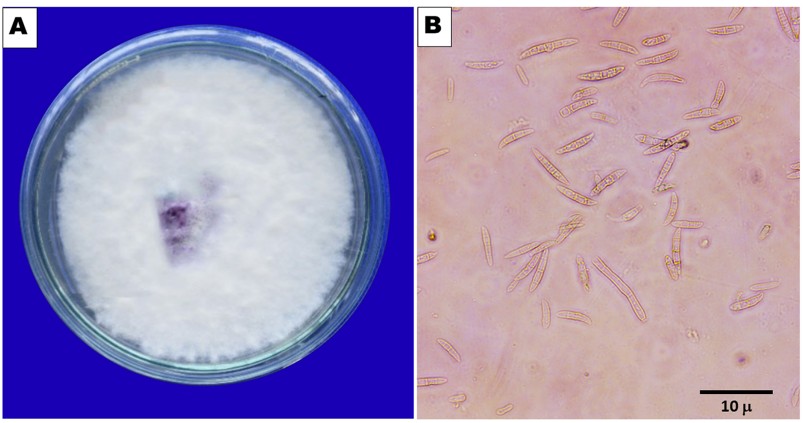

**Figure 2** (A) Pure culture of *F. oxysporum*. (B) Macro and microcondia (40 X).

**Figure 3** Pure cultures of five selected rhizobacterial isolates.

not utilized by both isolates. Glycerol and salicin as carbon sources were utilized by bacterial Iso 32 and mannitol by Iso 24, remaining carbon sources were not utilized.

The biochemical characterization of both isolates, Iso 32 and Iso 24 (Table 2), was analyzed using the online program for bacterial identification, "Advanced Bacterial Identification Software (ABIS)." The online identification results indicated 84.9% similarity with *Bacillus* spp. for Iso 32, while Iso 24 exhibited 78% similarity with *Bacillus* spp.

*Molecular characterization*

The 16S rRNA gene region was amplified using universal primers. PCR products were analyzed by gel electrophoresis and visualized under a gel documentation unit.

**Table 1 Carbohydrate utilization and biochemical properties of PGPR strains.**

| S. No | Test | Iso 24 | Iso 32 |
|---|---|---|---|
| **HiCarbo Kit, part C** | | | |
| 01 | Rhamnose | + | − |
| 02 | Cellobiose | − | − |
| 03 | Melezitose | − | − |
| 04 | α–Methyl-D-mannoside | − | − |
| 05 | Xylitol | − | − |
| 06 | ONPG | + | + |
| 07 | Esculin | + | + |
| 08 | D-Arabinose | − | − |
| 09 | Citrate | + | + |
| 10 | Malonate | + | + |
| 11 | Sorbose | − | − |
| 12 | Control | − | − |
| **HiCarbo Kit, partB** | | | |
| 01 | Inulin | − | − |
| 02 | Sodium gluconate | − | − |
| 03 | Glycerol | − | + |
| 04 | Salicin | − | + |
| 05 | Dulcitol | − | − |
| 06 | Inositol | − | − |
| 07 | Sorbitol | − | − |
| 08 | Mannitol | + | − |
| 09 | Adonitol | − | − |
| 10 | Arabitol | − | − |
| 11 | Erythritol | − | − |
| 12 | α–Methyl-D-glucoside | − | − |

Subsequently, the PCR products were sent for sequencing through outsourcing (Eurofins, Bangalore, Luxembourg). The obtained sequences were queried through a BLAST search (www.ncbi.nlm.nih.gov) and revealed a 99 to 100 percent homology with the already existing *B. amyloliquefaciens* sequences in GenBank. The sequence was deposited in the NCBI GenBank with accession number (MH491049). To establish the evolutionary relationship of the newly generated *B. amyloliquefaciens* isolates in this study, the existing 16S rRNA sequences of *B. amyloliquefaciens* were retrieved from the NCBI GenBank and utilized for phylogenetic analysis. The evolutionary history was inferred using the Neighbor-Joining method with 1,000 bootstrap replications, producing a similar tree topology forming relevant clades (Fig. 4). The analysis indicated that the Iso 32 isolate, isolated from the chilli rhizosphere, is phylogenetically similar to already reported *B. amyloliquefaciens* isolates around the world.

**Table 2 Biochemical identification of Iso 24 and Iso 32 based on ABIS online programme or bacterial identification.**

| Bacterial species | Similarity (%) | |
| --- | --- | --- |
| | Isolate 32 | Isolate 24 |
| *Bacillus mycoides* | 84.9% | – |
| *Paenibacillus residui* | 84.4% | – |
| *Paenibacillus alvei* | 79.3% | – |
| *Bacillus cereus* | 78.2% | – |
| *Bacillus galactosidilyticus* | – | 78% |
| *B. smithii* | – | 76.0% |
| *B megaterium* | – | 72.8% |
| *Brevibacillusb brevis* | – | 72.8% |

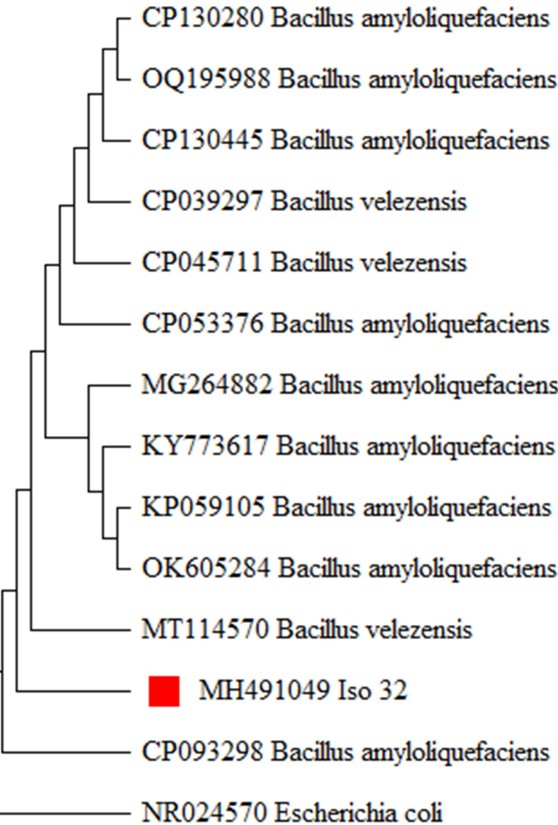

**Figure 4 Molecular phylogeny of Iso 32 (Sample 32) by maximum likelihood method.**

## HCN production

Two bacterial isolates *viz.*, Iso 24 and Iso 32 were screened for HCN production on KB agar medium. The Iso 24 did not show any colour change of Whatman no. 1 filter paper but the Iso 32 showed the colour change of the filter paper from deep yellow to reddish-brown colour indicating HCN production by the PGPR isolate (Iso 32).

**Table 3  Antagonistic activity of rhizobacterial isolates against *F. oxysporum* in dual culture.**

| Treatments | Radial growth (mm)* | Percent inhibition over control* | Inhibition zone (mm)* |
|---|---|---|---|
| Iso 01 | 27.0 | 70.00 (56.79)** | 14.3 |
| Iso 17 | 32.0 | 64.44 (53.40) | 13.0 |
| Iso 23 | 34.0 | 62.22 (52.07) | 13.3 |
| Iso 24 | 25.7 | 71.37 (57.65) | 18.7 |
| Iso 32 | 24.0 | 73.33 (58.91) | 21.7 |
| Control | 90.0 | 0.0 | 0.0 |
| SE (m) ± | 1.1 | | 0.5 |
| CD ≤ (0.05) | 3.4 | | 1.7 |

**Notes:**
\* Mean of three replications.
\*\* Values in the parenthesis are arcsine transformed values.

## Phosphate solubilization

The selected rhizobacterial isolates (Iso 24 and Iso 32) when subjected to phosphate solubilization in PKV media showed a varied range of phosphate solubilization zones. Bacterial strain Iso 24 demonstrated the highest phosphate solubilization zone (7 mm), surpassed by Iso 32 with a zone of 3.83 mm. Iso 32 exhibited the highest percentage of P-solubilization efficiency (PSE) with 143.75%, whereas Iso 24 followed closely with 116.67%. Additionally, Iso 32 recorded the highest P-solubilization index (PSI) with 1.44, while Iso 24 had 1.17.

## *In vitro* evaluation of selected rhizobacterial isolates against *Fusarium* wilt

Five selected bacterial isolates were discovered to be highly antagonistic to *F. oxysporum* f. sp. *capsici*. The percentage of disease inhibition against control varied from 73.3 to 62.2 percent (Table 3). Iso 32 (73.3%) (Fig. 5A) had the highest percentage of inhibition compared to the control, followed by Iso 24 (71.5%) (Fig. 5B). The Iso 32 (24 mm) and Iso 24 (25.76 mm) had the lowest radial growth, which was significantly different from the control's (90 mm) measurement.

## Evaluation of antagonistic rhizobacteria against *Fusarium* wilt using seed treatment and seedling root dip techniques

The impact of seed treatment with rhizobacterial isolates on seed germination in artificially inoculated protrays was assessed. Chilli seeds treated with five rhizobacterial isolates varied in percentage of germination from 84% to 92%, compared to 23% germination in control (Fig. 6). Iso 32 had the greatest germination (92%) of the individual isolates, followed by Iso 24 (88%). Some antagonistic effects were investigated for their potential to function as biocontrol agents against the wilt disease. As compared to the inoculated control using the seedling root dip technique, all the isolates significantly reduced the wilt disease. Using Iso 32, which exhibited 85.5% disease control above inoculated control, the incidence of the disease decreased to a level of 8.83% (Table 4); (Fig. 7).

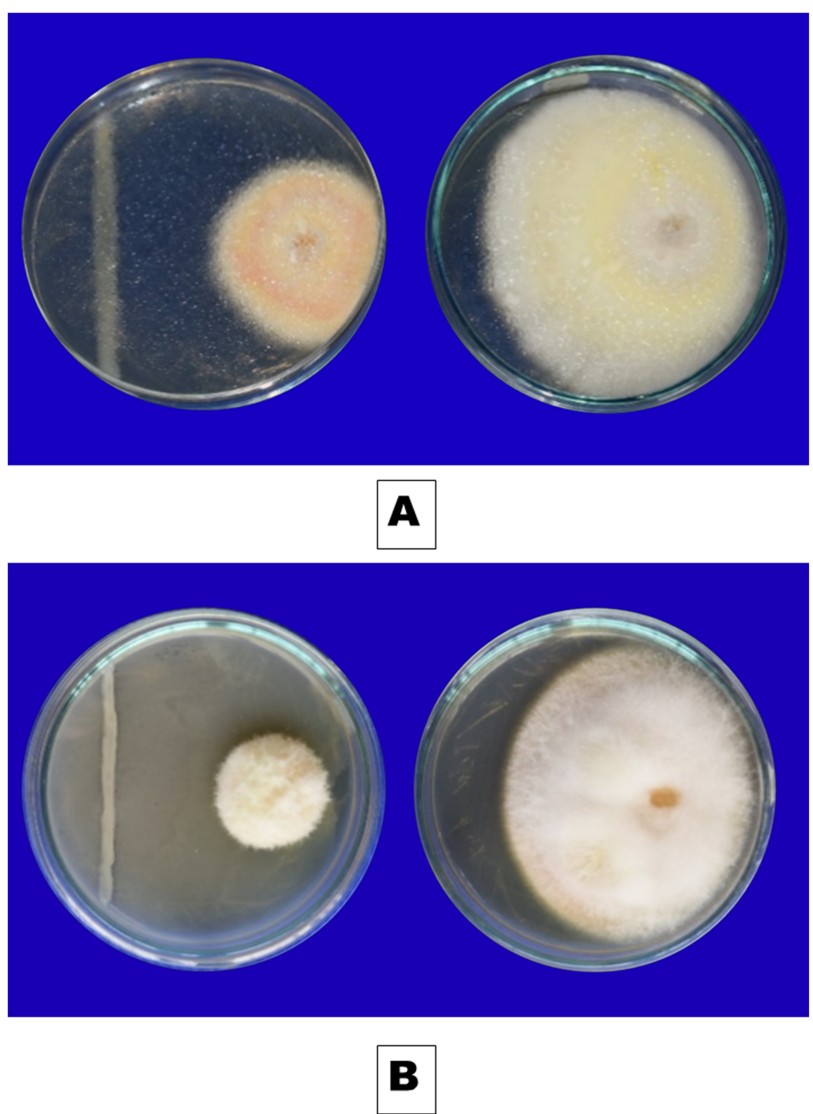

**Figure 5** Inhibitory effect of rhizobacterial isolates (A) Iso 32, (B) Iso 24, on *F. oxysporum* using dual culture technique.              

## Effectiveness of rhizobacterial treatment using rolled paper towel and root dip methods on plant growth parameters

Five isolates were chosen and thoroughly investigated for plant growth response through *in vitro* seed treatment using the rolled towel method and the root dip method under greenhouse conditions, based on the screening for antagonistic potential revealed through the dual culture technique and other characteristics.

### Rolled paper towel method

The effects of treating seeds with rhizobacterial isolates on germination, root length, shoot length, and vigour index were observed (Table 5). A notable difference in germination rates was observed, ranging from 96.67% to 87.33%. However, seeds treated with Iso 32 exhibited the maximum germination rate of 96.67%, whereas the control group of

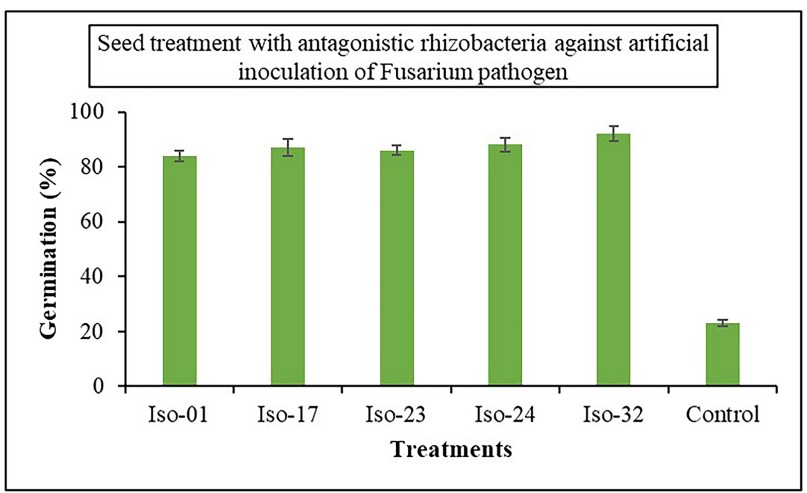

**Figure 6 Effect of seed treatment with native rhizobacterial isolates against artificially inoculated fusarium pathogen on germination (%) under *in vivo* conditions.**

**Table 4 Effect of seed treatment and seedling root dip treatment with native rhizobacterial isolates on incidence of wilt under artificial inoculation of pathogen.**

| Treatments | Germination (%) | Percent disease incidence (PDI) | Disease reduction over Control (%) |
|---|---|---|---|
| Iso 01 | 84.00 (66.42)* | 12.77 (20.93)* | 79.07 (62.78)* |
| Iso 17 | 87.00 (68.87) | 13.83 (21.83) | 77.32 (61.56) |
| Iso 23 | 86.00 (68.03) | 12.90 (21.05) | 78.85 (62.62) |
| Iso 24 | 88.00 (69.73) | 10.20 (18.63) | 83.28 (65.86) |
| Iso 32 | 92.00 (73.57) | 8.83 (17.29) | 85.6 (67.63) |
| Control | 23.00 (28.66) | 61.00 (51.35) | 00.0 |
| SE (m) ± | 1.5 | 0.72 | |
| CD (≤ 0.05) | 4.6 | 2.24 | |

**Note:**
* Values in the parenthesis are arcsine transformed values.

untreated seeds showed an 86% germination rate (Fig. 8). When seeds were treated with various rhizobacterial isolates, root length, shoot length, and seedling vigour index significantly varied. Compared to the minimal shoot length (3.72 cm) in the control, isolate 32 had a maximum shoot length of 5.60 cm. As opposed to the control seedlings' (3.54 cm) root length, isolate-32′ s maximum root length (5.49 cm) was measured. The highest vigour index (1071.71) could be achieved by the seedlings treated with Isolate 32, closely followed by the vigour index (964.35) in seedlings treated with Iso 24 compared to the control (624.07) (Fig. 8).

### *Root dip method*

Results of a pot experiment carried out in a greenhouse to assess the *in vivo* impact of rhizobacterial isolates treated with a root dip on plant growth parameters including shoot length, root length, and fresh and dry weight (Fig. 9). Evaluation of the effect of root dip

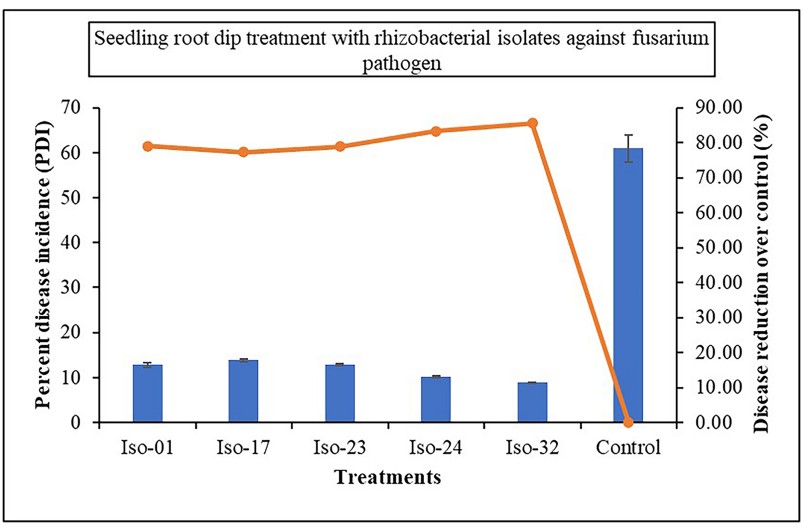

**Figure 7 Effect of seedling root dip treatment with native rhizobacterial isolates against artificially inoculated fusarium pathogen on percent disease incidence (PDI) under *in vivo* condition.**

**Table 5 Effect of seed treatment with rhizobacterial isolates on seedling growth parameters by rolled paper towel method.**

| Treatments | Germination % | Root length (cm) | Shoot length(cm) | Vigour index |
|---|---|---|---|---|
| Iso 01 | 90.00 (71.57)* | 3.78 | 4.65 | 758.20 |
| Iso 17 | 91.67 (73.22) | 4.06 | 4.28 | 772.56 |
| Iso 23 | 87.33 (69.15) | 4.62 | 5.33 | 884.40 |
| Iso 24 | 92.67 (74.29) | 5.01 | 5.27 | 948.80 |
| Iso 32 | 96.67 (79.48) | 5.04 | 5.07 | 973.70 |
| Control | 86.00 (68.03) | 3.55 | 3.85 | 636.48 |
| SE (m) ± | 2.06 | 0.11 | 0.10 | 25.94 |
| CD (≤ 0.05) | 6.18 | 0.33 | 0.32 | 77.67 |

**Note:**
 * Values in the parenthesis are arcsine transformed values.

treatment of rhizobacterial isolates on plant growth parameters such as shoot length, root length, and fresh and dry weight (Table 6). Iso 32 treatment produced the longest root length (14.75 cm), whereas Iso 17 treatment produced the shortest root length (11.43 cm). Similarly, the Iso 32 had the highest fresh root weight (2.65 g). Likely substantially greater than the control, Iso 32 (1.41 g) and Iso 01 (0.84 g) showed the highest root dry weight (0.36 g). In case of shoot parameters, Iso 01, had the longest shoot length (14.23 cm), highest fresh weight (3.38 g) and dry weight (2.11 g) respectively followed by Iso 32 had superior characters. Which was considerably greater than the control shoot length (9.33 cm), fresh weight (0.76 g) and dry weight (0.54 g) respectively. In terms of statistics, all of the treatments were comparable.

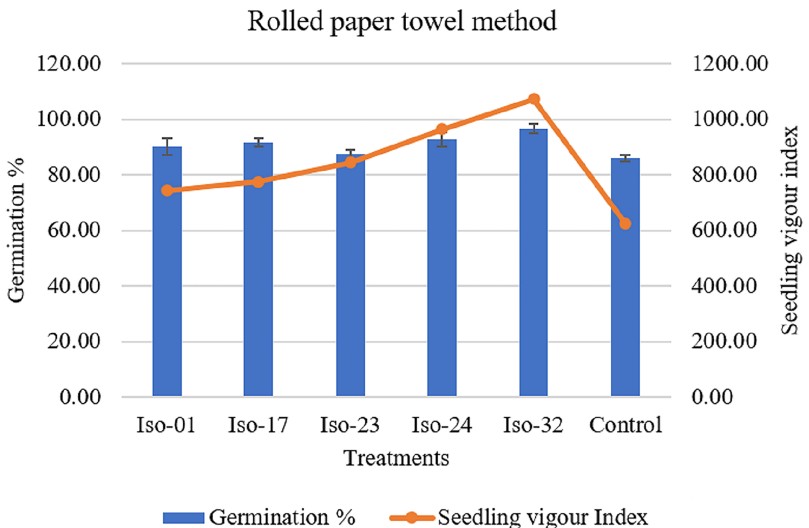

**Figure 8** Effect of rhizobacterial isolates on seed germination and seedling vigour under rolled paper towel method.

**Figure 9** Seedling growth parameters of chilli crop under *in vivo* conditions.

## DISCUSSION

Native PGPR has a very crucial role in augmenting plant growth and suppressing disease-causing agents, which eventually enhances growth and yield (*Compant et al., 2005*). The present focus of the study was to inhibit the *Fusarium* wilt by exploiting native antagonists and PGPR so that it could be used effectively in the future. Biological control by using antagonistic native rhizobacteria is a viable source to suppress the *Fusarium* wilt since they do not pose a threat to human health or the environment. Therefore, in this way, the incidence of *Fusarium* wilt can be reduced in chilli. The rhizosphere is rich in nutrient content due to the accumulation of root exudates and metabolites of diverse origins; it harbours a diverse population of rhizobacteria (*Yang et al., 2008*). So, the isolation of *Bacillus* spp. strain (Iso 32) was carried out from the rhizosphere of healthy chilli plants.

After the isolation of fifty-five rhizobacterial strains from chilli plants, five strains, namely isolates Iso 01, Iso 17, Iso 23, Iso 24, and Iso 32, were found to efficiently suppress the growth of *F. oxysporum* f. sp. *capsici* in preliminary screening with more than 55%. Out of those five strains, Iso 32 (73.3%) and Iso 24 (71.5%) caused the highest levels of pathogen inhibition. Under greenhouse trials, artificially pathogen-inoculated chilli plants treated with Iso 32 (8.8%) and Iso 24 (10.2%) had decreased PDI, with a percent disease reduction over control of 85.6% and 83.3%, respectively (Table 1). Therefore, native rhizospheric antagonistic bacteria in this study were found effective in inhibiting the growth of *F. oxysporum* f. sp. *capsici* under *in vitro* and *in vivo* conditions. Our experiment results were correlated with previous studies in which *Bacillus* spp. showed antagonistic activity against pathogens (*Chowdhury, Majumdar & Mandal, 2020*; *Delgado-Ramírez, Hernández-Martínez & Sepúlveda, 2021*; *Dowarah et al., 2021*).

Cultural characterization of all five isolates was carried out, and it was observed that after 7 days of incubation, there were variable growth patterns (Table 2). Similarly, carbohydrate utilization tests (Himedia kits) were carried out with Iso 24 and Iso 32, as they showed superior antagonistic activity compared to other strains (Table 3). Both isolates showed variable responses for specific breakdown products, showing that they are different from each other. Through 16S rRNA gene sequence studies, it was proven that Iso 32 has a stronger genetic affinity for *Bacillus* spp. Similar results were reported by *Kumar & Audipud (2014)*, that rhizobacteria isolated from chilli plants were identified as *Bacillus* spp. on the basis of 16S rRNA partial gene sequence analysis.

All the isolates were tested for phosphate solubilization capacity in PKV media. A variable range of phosphate solubilization zones was observed in all the isolates. Similar results were reported by *Tallapragada & Seshachala (2012)*, who studied *Aspergillus*, Actinomycetes, *Bacillus, Nocardia, Streptomyces*, and certain yeasts for phosphate solubilization capacity. HCN production by rhizobacteria plays an important role in inhibiting the pathogen. Iso 32 was found to be a strong HCN producer. HCN production by antagonistic bacteria has been reported by other workers as well (*El-Sersawy et al., 2021*; *Kashyap et al., 2021*; *Sehrawat, Sindhu & Glick, 2022*).

Under greenhouse conditions, studies revealed that the seeds treated with rhizobacterial strains showed higher seed germination and a lower disease incidence when compared to

**Table 6 Efficacy of potential antagonistic bacteria on enhancing seedling growth parameters of chilli crop under *in vivo* conditions.**

| Treatment | Root | | | Shoot | | |
|---|---|---|---|---|---|---|
| | Length (cm) | Fresh weight (gm) | Dry weight (gm) | Length (cm) | Fresh weight (cm) | Dry weight (cm) |
| Iso 01 | 13.60 | 0.97 | 0.52 | 13.98 | 1.71 | 0.83 |
| Iso 17 | 12.88 | 1.51 | 0.67 | 13.08 | 2.73 | 1.26 |
| Iso 23 | 13.40 | 1.17 | 0.62 | 12.93 | 1.88 | 0.98 |
| Iso 24 | 10.65 | 1.580 | 0.80 | 15.00 | 1.96 | 0.94 |
| Iso 32 | 14.93 | 2.11 | 1.04 | 16.85 | 2.72 | 1.23 |
| Control | 9.68 | 0.79 | 0.399 | 10.19 | 1.47 | 0.66 |
| SE (m) ± | 0.97 | 0.07 | 0.05 | 0.11 | 0.07 | 0.06 |
| C. D. (≤0.05) | 0.31 | 0.23 | 0.17 | 0.35 | 0.09 | 0.19 |

the control. Iso 32 had the greatest germination (92%) of the individual isolates, followed by Iso 24 (88%). Some antagonistic effects were investigated for their potential to function as biocontrol agents against the wilt disease. Similar results were reported by *Mohan (2006)* that seed treatment of tomatoes reduces the damping off disease caused by *Fusarium* sp. Iso 32, which provided 85.6% disease control above inoculated control, decreased the incidence of the disease to a level of 8.8% in the seedling root dip technique. *Bacillus subtilis* and *Pseudomonas fluorescens* were the best strains for controlling *Sclerotium rolfsii* and *F. oxysporum*, respectively. These results were reported by *Zaghloul et al. (2008)*.

The plant growth promotion activity of rhizobacterial strains under lab conditions by the roll towel method was proved to be effective in germination, shoot length, and root length with Iso 32 (Table 6). A pot experiment carried out under greenhouse conditions for plant parameters and biomass proved to be superior with Iso 32 when compared to other isolates. Similar results were reported by *Maiyappan et al. (2010)*. *Bacillus* spp. has been exploited for its plant growth-promoting efficacy. *Malleswari, Bagyanarayana & Rana (2015)* also reported that rhizobacterial isolates increased various growth parameters.

## CONCLUSION

The chilli crop is heavily reliant on pesticides for disease control. However, there is a growing need for sustainable solutions as the edible parts of the plants come into direct contact with toxic chemicals. Native microbes, such as rhizobacterial strains from the chilli crop, can effectively treat Fusarium wilt and promote plant development. The current research concluded that native rhizobacterial strains obtained from the chilli rhizosphere might be effectively employed for treating plants for *Fusarium* wilt and promoting plant development. Two of the five rhizobacterial isolates, Iso 32 and Iso 24, have shown promising antagonistic abilities and promoted plant growth, paving the way for environmentally friendly wilt disease control. Further investigation is needed to understand the precise mechanisms of these isolates' control and development.

## ACKNOWLEDGEMENTS

The authors thank the Head of Central Horticultural Experiment Station, Bhubaneswar (Regional research station of ICAR-IIHR, Bengaluru) for providing facilities for carrying out research.

### Funding

The authors received no funding for this work.

### Competing Interests

Ravinder Kumar is an Academic Editor for PeerJ.

### Author Contributions

- Bhanothu Shiva conceived and designed the experiments, analyzed the data, prepared figures and/or tables, authored or reviewed drafts of the article, and approved the final draft.
- Petikam Srinivas conceived and designed the experiments, analyzed the data, prepared figures and/or tables, authored or reviewed drafts of the article, and approved the final draft.
- Deepa Khulbe conceived and designed the experiments, performed the experiments, prepared figures and/or tables, and approved the final draft.
- Lellapalli Rithesh conceived and designed the experiments, performed the experiments, prepared figures and/or tables, authored or reviewed drafts of the article, and approved the final draft.
- Penumatsa Kishore Varma conceived and designed the experiments, performed the experiments, authored or reviewed drafts of the article, and approved the final draft.
- Rahul Kumar Tiwari conceived and designed the experiments, performed the experiments, analyzed the data, authored or reviewed drafts of the article, and approved the final draft.
- Milan Kumar Lal conceived and designed the experiments, analyzed the data, prepared figures and/or tables, authored or reviewed drafts of the article, and approved the final draft.
- Ravinder Kumar conceived and designed the experiments, analyzed the data, prepared figures and/or tables, authored or reviewed drafts of the article, and approved the final draft.

### DNA Deposition

The following information was supplied regarding the deposition of DNA sequences:

The sequences are available at NCBI: MH491049.1.

### Data Availability

The raw data is available in the Supplemental File.

## Supplemental Information

Supplemental information for this article can be found online at http://dx.doi.org/10.7717/peerj.17578#supplemental-information.

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
