# Peer review of "Isolation and characterization of native antagonistic rhizobacteria against Fusarium wilt of chilli to promote plant growth"

_PeerJ, doi:10.7717/peerj.17578_

## Round 0.1 · original submission · Major Revisions

Kindly revise your manuscript thoroughly. The methods and experimental design should be revised. Follow the reviewers comments and mention if the experiments were done in replicates and the methods used are mentioned clearly and properly cited.

Reviewer 1 ·

Basic reporting

The manuscript titled "Isolation and characterization of native antagonistic and plant growth promoting rhizobacteria against Fusarium Wilt of Chilli" by Bhanothu Shiva and colleagues investigates the severe issue of Fusarium wilt disease in chillies. This disease is particularly hard to control with chemical fungicides, given its soil-borne nature. The research draws from the natural rhizosphere soil of the chilli plant to isolate bacterial antagonists and evaluate their capacity to both antagonize the Fusarium pathogen and promote plant growth. From fifty-five strains isolated, five were identified as notably antagonistic against Fusarium in vitro. Particularly, Iso 32 and Iso 24 demonstrated high levels of pathogen inhibition and markedly reduced disease incidence in greenhouse trials. These isolates also augmented the seed vigor index and enhanced other plant growth metrics. Further analyses revealed that Iso 32 was molecularly characterized and identified as Bacillus spp. (MH491049). The results suggest that these native rhizobacteria can protect chilli plants against Fusarium infection while also fostering their overall growth.
The introduction should include more details and examples. And in line 80, “in this study”, it will be better if the author could summarize more of their research at the end the Introduction. The author should use high resolution photos and should not change the original ratio of photo height/width. The results part should focus on more of the experimental observation and conclusion rather than giving experimental details. The statistical analysis is also weak.

Experimental design

There is lacking proof for how Iso24 and Iso32 are superior over other isolates.
Do all the experiments have replicates? Almost no information has been given. How many technical and biological replicates for each experiment?
The photos for figure 7 should be taken from the exact same angel, also should put a ruler to set the same scale.

Validity of the findings

The author only mentioned in the discussion that in total 55 rhizobacterial strains from chili plants were isolated, should include the data how 5 most effective isolates were picked. Also it’s better to provide photos showing the morphology of all 5 isolates.
Figures for the results of HCN production should be presented.
Many characterizations are only for Iso24 and Iso32, why suddenly the author provide the result of phosphate solubilization for 8 isolates?
The ruler in figure 8 is not-readable, also I suggest the author to add plot figures for all the plant growth data.

Additional comments

1. I suggest rephrasing the title as “Isolation and characterization of native antagonistic rhizobacteria against Fusarium Wilt of Chilli to promote plant growth”.
2. Line 35: levels should be level.
3. Line 50: Bacillus: Wilt should be Bacillus; Wilt
4. The format of all the references should be corrected according to PeerJ Instructions for Author.
5. Line 63: Please explain the part “emend. Synd. & Hans.”?
6. I suggest to add the location details (latitude and longitude) for sampling in the Materials & Methods.
7. Line 150: Please explain 28 ± 20 °C ?
8. Line 196: I suggest “from the wilted chili plant samples”
9. All figure should be cited as Fig. x according to PeerJ Instructions for Author.
10. Figure 1: Suggest adding portrait photos showing the velvety Fusarium. Eg. On leaves and on fruits.
11. Figure 2: title spelled wrong.
12. Line 246: missing .
13. Figure 3 and figure 4 are cited in the main text in a wrong order. The photo is blurry and disproportioned. Two backgrounds look different for two isolates. Missing correct labels.
14. Figure 4: why is the corresponding taxonmic clustering percentage for sample 32 and Bacillus spp. so low as 16?
15. No other species identified for Iso24 other than Bacillus galactosidilyticus? No tree?
16. Line 253: Please rephrase this sentence to improve grammar and calrity.
17. Line 286-287: please check the sentence.

Reviewer 2 ·

Basic reporting

Please see the attached file where I added my comments.

Experimental design

Please see the attached file where I added my comments.

Validity of the findings

Please see the attached file where I added my comments.

Additional comments

Please see the attached file where I added my comments.

Annotated reviews are not available for download in order to protect the identity of reviewers who chose to remain anonymous.

Reviewer 3 ·

Basic reporting

1. Recent related references may be added in the Manuscript

Experimental design

1. It is up to the mark.
2. Methods should be described in details as mentioned in the text lines 93 - 96; 106- 108, and185- 192 with sufficient information.

Validity of the findings

1. This part is required to strengthen by rewriting the finding as suggested in lines 200- 205.
2. Revise the finding of 3.2.1 molecular characterization.

Annotated reviews are not available for download in order to protect the identity of reviewers who chose to remain anonymous.

---

## Round 0.2 · Major Revisions

Thanks for your revised version of your manuscript. Unfortunately, there are still major revisions required for the final acceptance and suitability for publication.
Kindly check the reviwer’s attached comments and revise your manuscript accordingly.

Reviewer 1 ·

Basic reporting

The author has made several improvements based on the reviewer's comments: The introduction has been expanded to include more details and examples. Additional evidence and analysis are provided to support the claim that certain isolates are superior to others. Also, photos displaying the morphology of all five isolates have been included in the Results section and figures depicting the results of HCN production and plot figures for all plant growth data have also been incorporated.
Overall, these revisions significantly improved the manuscript. However, the resolution for Figure 1, 2, 4, 5 and 9 is still low quality. Figure 2 and 5 are not in their original proportions and have been distorted. For figure 2B, the scale bar should be added for the microscope image and for figure 9, the electronic scale bar should be added and all the photos should be adjusted to the same scale and also in their original proportions.

Experimental design

NA

Validity of the findings

NA

Additional comments

NA

---

## Round 0.3 · accepted · Accept

Thanks for revising your manuscript and modify according to the reviewer's comments. I am pleased to inform you that it is suitable for publication in its current form.

Reviewer 1 ·

Basic reporting

Based on revision of manuscript, authors successfully incorporated the point to point wise through the manuscript as suggested comments so that I would like to strongly recommend that acceptance of manuscript in your prestigious journal for publication.

Experimental design

NA

Validity of the findings

NA

Additional comments

NA